

# How a European network may help estimating methane emissions at the French national scale

Isabelle Pison[1], Antoine Berchet[2], Marielle Saunois[1], Philippe Bousquet[1], Grégoire Broquet[1], Sébastien Conil[3], Marc Delmotte[1], Anita Ganesan[5], Olivier Laurent[1], Damien Martin[4], Simon O'Doherty[5], Michel Ramonet[1], T. Gerard Spain[6], Alex Vermeulen[7,a], and Camille Yver Kwok[1]

[1]Laboratoire des Sciences du Climat et de l'Environnement, LSCE-IPSL (CEA-CNRS-UVSQ), Université Paris-Saclay, 91191 Gif-sur-Yvette, France.
[2]Laboratory for Air Pollution/Environmental Technology, Empa - Swiss Federal Laboratories for Materials Science and Technology, Dübendorf, Switzerland
[3]Agence Nationale pour la gestion des Déchets RadioActifs, Châtenay-Malabry, France
[4]Centre for Climate and Air Pollution Studies, School of Physics, National University of Ireland Galway, Galway, Ireland
[5]Atmospheric Chemistry Research Group, School of Chemistry, University of Bristol, Cantocks Close, Bristol, United Kingdom
[6]National University of Ireland Galway, Galway, Ireland
[7]Energy Research Centre of the Netherlands, Heerhugowaard, The Netherlands
[a]now at Dept. Phys. Geography and Ecosystem Science, Lund, Sweden

*Correspondence to:* I. Pison (isabelle.pison@lsce.ipsl.fr)

**Abstract.** Methane emissions at the national scale in France in 2012 are inferred by assimilating continuous atmospheric mixing ratio measurements from nine stations of the European network ICOS located in France and surrounding countries. To assess the robustness of the fluxes deduced by our inversion system based on an objectified quantification of uncertainties, two complementary inversion set-ups are computed and analysed: i) a regional run correcting for the spatial distribution of

5   fluxes in France, and ii) a sectorial run correcting fluxes for activity sectors at the national scale. In addition, our results for the two set-ups are compared with fluxes produced in the framework of the inversion inter-comparison exercise of the InGOS project. The seasonal variability of fluxes is consistent between different set-ups, with maximum emissions in summer, likely due to agricultural activity. However, very high monthly posterior uncertainties (up to ≈65% to 74% in the sectorial run in May and June) makes it difficult to attribute maximum emissions to a specific sector. At the yearly national scale, the two

10   inversions range to 3835–4050 GgCH$_4$ and 3570–4190 GgCH$_4$ for the regional and sectorial run, respectively, consistently with the InGOS products. These estimates are 25 to 55% higher than the total national emissions from bottom-up approaches (biogeochemical models from natural emissions, plus inventories for anthropogenic ones), consistently pointing at missing or under-estimated sources in the inventories and/or in natural sources. More specifically, in the sectorial set-up, agricultural emissions are inferred as 66% larger than estimates reported to UNFCCC. Uncertainties on the total annual national budget are

15   108 GgCH$_4$ and 312 GgCH$_4$, i.e, 3 to 8%, for the regional and sectorial run respectively, smaller than uncertainties in available bottom-up products, proving the added value of top-down atmospheric inversions. Therefore, even though the surface network used in 2012 does not allow to fully constrain all regions in France accurately, a regional inversion set-ups makes it possible to provide estimates of French methane fluxes with an uncertainty on the total budget less than 10% at the yearly scale. Additional



sites deployed since 2012 would help to constrain French emissions at finer spatial and temporal scales and attributing missing emissions to specific sectors.

# 1 Introduction

Methane ($CH_4$) is the second most important anthropogenic greenhouse gas in terms of impact on climate change (after $CO_2$),
due to its global warming potential 28 times larger than that of $CO_2$ over a 100 year period (IPCC, 2014), and possibly even larger (Holmes et al., 2013). Consequently, it is a very good candidate for climate change mitigation policies.

$CH_4$ is emitted by a variety of sources. Most $CH_4$ sources ($\approx$60% in mass) are linked to microbial activity in anaerobic environments: mainly natural wetlands, anthropogenically managed wetlands (such as rice-paddies), landfills, waste-water facilities and the intestines of wild and domesticated animals. $CH_4$ is also emitted from fossil fuel related processes, through natural geologic gas seeps or during the exploitation and distribution of gas, oil and coal. Finally, $CH_4$ is emitted by biomass burning, through incomplete combustion, mainly in wild fires, biomass burning due to agricultural activities and the use of biofuels. This variety of sources and the strong spatial and temporal heterogeneity of emissions lead to uncertainties on $CH_4$ global and regional budgets, which remain large enough to impair our understanding of atmospheric variations of $CH_4$ concentrations and particularly, the attribution of $CH_4$ mixing ratio variations to specific sources and/or zones (Saunois et al., 2016, 2017).

$CH_4$ emissions are reported yearly to the UNFCCC (United Nations Framework Convention on Climate Change) by the countries that are parties to the convention, both in the framework of the convention and of the Kyoto protocol. Reporting $CH_4$ emissions at the national scale to the UNFCCC is currently done by bottom-up approaches, which include inventories (mainly for anthropogenic emissions) and biogeochemical models (mainly for anthropogenic emissions due to biogenic processes and natural emissions). For instance, French methane emissions represent about 13% of EU-28 ones (according to UNFCCC 2012 data) and are reported by the CITEPA (Centre Interprofessionnel Technique d'Études de la Pollution Atmosphérique), an institute that compiles inventories. Inventories are based on collecting and aggregating huge amounts of data and information (e.g., activity statistics, emission factors). The IPCC (2006) provides guidelines to build inventories for reporting to the UNFCCC, classifying the methodologies in 3 tiers, from the simplest to implement (Tier 1, which uses default activities and emission factors provided by IPCC) to the most complex (Tier 3, which may include models and is supposed to lead to smaller uncertainties). The Tier 1 uncertainty is the most straightforward to obtain since it combines the uncertainties on the activity and the emission factor. From these guidelines, the CITEPA provides annual emissions of $CH_4$ in mainland France for anthropogenic activity sectors together with Tier 1 uncertainties for the major contributing sectors, ranging from 16% ($\approx \pm$212 Gg $CH_4$ in 2012) for enteric fermentation to 104% ($\approx \pm$90 Gg $CH_4$ in 2012) for waste water treatment and discharge (CITEPA, 2016). In October 2016, another French inventory was released. This inventory, called Inventaire National Spatialisé (Ministère de l'Environnement, de l'Énergie et de la Mer, 2017), provides emissions at a kilometric resolution; for the year 2012, the kilometric maps are based on the CITEPA's national totals. $CH_4$ anthropogenic emissions for France are also provided by larger scale inventories: IER (Pregger et al., 2007) at the European scale and four inventories covering the whole world: EDGAR

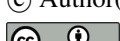



(Janssens-Maenhout et al., 2017), ECLIPSE (Stohl et al., 2015), EPA (EPA, 2012), FAO (FAOSTAT: Food and Agriculture Organization of the United Nations, 2017). For natural $CH_4$ emissions in France, we use here the emissions provided by bio-geochemical models at the global scale for wetlands and termites in the context of Saunois et al. (2016). The difficulties of bottom-up approaches are mainly due to missing information. For example, inventories may miss either statistics on activity sectors or even sources. Moreover, inventory uncertainties remain high, for instance, at the national scale due to errors in the aggregation of statistical information or to uncertainties on the emission factors. Also, inventories do not often associate uncertainties to their estimates. For UNFCCC reporting, the CITEPA provides uncertainties for the main emitting sectors in France: the uncertainty on French anthropogenic $CH_4$ emissions in 2012 is then at least $\pm 26\%$.

In this context, top-down approaches may help bringing more information to emissions estimated by inventories. Top-down approaches are based on the assimilation of atmospheric data (in our case, measurements of atmospheric mixing ratios) into a chemistry-transport model using prior information on the emissions. Within an inverse-modelling framework, the data, model and prior emissions, together with their respective error statistics, are optimally combined to provide posterior emissions (with their own uncertainties, depending on the method used). The atmospheric signal integrates all emissions so that sources which are not explicitly described in bottom-up approaches are taken into account in top-down approaches. Top-down approaches are widely used at the global scale (for a review, see Saunois et al., 2016). Recent studies have also used top-down approaches at regional scales for large regions such as the Arctic (Thompson et al., 2017), Eurasia (Berchet et al., 2015b), East Asia (Thompson et al., 2015) or the USA (Jeong et al., 2016). These regional studies are either global with a zoom or focus over a specific region of interest or domain-limited at fine scales; almost all of the studies use surface data, sometimes with the addition of satellite data. Studies at national scales for countries about the size of France are not numerous. In European studies, the atmospheric measurement data are mostly provided from national and European surface networks: Henne et al. (2016) estimated the Swiss national total of $CH_4$ emissions; Ganesan et al. (2015) examined the $CH_4$ (and nitrous oxide) emissions in Ireland and the United Kingdom; Bergamaschi et al. (2015a) and Bergamaschi et al. (2017) analysed methane emissions in Europe at the regional or country scales, including France.

Although top-down studies are promising, their robustness is limited by *i)* the availability of observations, which must be numerous enough in time and well-distributed in space over the relatively small (compared to the global scale) area of interest, *ii)* for most of them, the lack of expert-knowledge for defining the set-up of the inverse system (i.e. prescribing the error statistics, including the spatial and temporal correlations in prior emissions, which may be assumed to be highly country-dependent), and *iii)* the issue of representing at best the atmospheric transport at this scale. It is indeed important to assess which spatio-temporal scales are actually constrained by the assimilated data in order to exploit as much information as possible while avoiding over-interpretation of the results (e.g., at too fine scales). This issue arises particularly when estimating emission budgets at the national scale in rather small countries, like France and most countries in Western Europe.

Studies aiming at estimating European greenhouse gas (GHG) emissions can take advantage of measurements from the ICOS (Integrated Carbon Observatory System) network. ICOS is a European research infrastructure, of which one of the main objectives is to quantify European GHG fluxes. To do so, a number of European national measurement networks cooperate to ensure the monitoring of GHG atmospheric concentrations and fluxes in terrestrial and marine ecosystems, as well as the





distribution of the data with a common high quality standardization. The ICOS network of atmospheric stations performs continuous in-situ measurements, made both from ground stations and tall towers.

This study aims at estimating $CH_4$ emissions in mainland France. We use an inversion framework that allows us to overcome the issue of prescribed error statistics. The data to assimilate are atmospheric measurements available from the ICOS network
in 2012. In particular, we aim at determining whether the current status/deployment of the ICOS network is sufficient to infer French methane emissions by answering the following questions. What constraints may such a network bring on French emissions at the national scale? What spatio-temporal scales are constrained in France, which is a country with large regional variations in emissions? Which characteristics of the French national budgets can be inferred: uncertainties, seasonal variations, types of processes? For example, is it possible to infer seasonal variations? Is the uncertainty on the total annual budget for
France smaller than the uncertainty on bottom-up inventories?

The methodological framework of our study is presented in Section 2, with a focus on the tools provided for the interpretation of the results (Section 2.3). The inversion set-ups used for inferring methane emissions in France are described in Section 3. Results are discussed in Section 4, first in terms of the relevancy of the features informed by the inversion (Section 4.1, Section 4.2) then in terms of French methane emissions (Section 4.3).

## 2  Inverse method

### 2.1  General inverse framework

In the framework of atmospheric inversion, the most common notations are the following: $x$ for the state vector, including the emission fluxes to be optimized at the chosen spatial and temporal scales; $x^b$ for the prior estimate of the state vector; $y^o$ for the observation vector, consisting here in $CH_4$ atmospheric concentration data. The observations and the prior state
are associated with their covariance error matrices $R$ and $P^b$ respectively. $R$ includes the errors on the measurements (e.g., instrumental errors) plus the errors on the transport in the model and on the representativity of the grid cell compared to the measurement. The link from the state vector to the observation space is made by the observation operator $H$. Here, $H$ represents the atmospheric transport and mixing on the model's grid and the space and time filtering of the simulated concentrations to obtain the equivalent of the observation data. Since the lifetime of $CH_4$ in the atmosphere is very long ($\approx$9 years) compared to
the residence time of air masses in the domain of interest in this study ($\approx$3-5 days), chemistry is not taken into account so that $H$ is assumed to be linear and its Jacobian $\mathbf{H}$ is used, with $H(x) = \mathbf{H}x$.

As mentioned previously, the inversion optimally combines the prior knowledge, the knowledge on which the model is based and the knowledge brought by the data to be assimilated: it consists of finding the probability density function (pdf) of the state $x$ knowing both the prior $x^b$ and the differences between the observations $y^o$ and their equivalents computed by the model
$H(x^b)$. For any possible state $x$, this probability is $p(x|y^o, x^b)$. To characterize $p(x|y^o, x^b)$, it is usual to use the Bayesian framework and to assume that uncertainties in the system follow Gaussian functions. As a result, the posterior state vector $x^a$



and its associated covariance matrix of posterior errors $\mathbf{P}^{\mathrm{a}}$ are given by:

$$\boldsymbol{x}^{\mathrm{a}} = \boldsymbol{x}^{\mathrm{b}} + \mathbf{K}(\boldsymbol{y}^{\mathrm{o}} - \mathbf{H}\boldsymbol{x}^{\mathrm{b}}) \tag{1}$$

$$\mathbf{P}^{\mathrm{a}} = \mathbf{P}^{\mathrm{b}} - \mathbf{KHP}^{\mathrm{b}} \tag{2}$$

with $\mathbf{K}$ the Kalman gain matrix, given by:

$$\mathbf{K} = \mathbf{P}^{\mathrm{b}}\mathbf{H}^{\mathrm{T}}(\mathbf{R} + \mathbf{HP}^{\mathrm{b}}\mathbf{H}^{\mathrm{T}})^{-1}. \tag{3}$$

If $\mathbf{R}$ and $\mathbf{P}^{\mathrm{b}}$ are given, the inversion is a direct computation from these formulae (providing the sizes of the matrices are adapted to the computing resources). As stated before, $\mathbf{R}$ and $\mathbf{P}^{\mathrm{b}}$ are generally derived from expert knowledge based on studies on the atmospheric transport, the performances of the models, etc. Such knowledge is quite established for the global scale and large region scales, but is not readily available yet for GHGs at the smaller national scales. Therefore, defining $\mathbf{R}$ and $\mathbf{P}^{\mathrm{b}}$ is not an easy task at the country scale (scale of interest here) while mis-specifying $\mathbf{R}$ and $\mathbf{P}^{\mathrm{b}}$, and more especially their relative weights, has a very strong impact on the results of the inversion.

### 2.2 Principle and main steps of the marginalized Bayesian inversion method

In order to avoid multiple tests on the structures and values of $\mathbf{R}$ and $\mathbf{P}^{\mathrm{b}}$, we use here the marginalized Bayesian inversion method, which is an extension of the classical Bayesian inversion framework, developed and implemented by Berchet et al. (2015a). Instead of classically inferring the posterior state $\boldsymbol{x}^{\mathrm{a}}$ and its covariance matrix $\mathbf{P}^{\mathrm{a}}$ directly from prescribed prior uncertainties in the covariance matrices $\mathbf{R}$ and $\mathbf{P}^{\mathrm{b}}$, the method uses a sample of the continuous distribution of all the possible couples of prior uncertainties $(\mathbf{R}, \mathbf{P}^{\mathrm{b}})_i$ to produce an ensemble of the posterior counterparts $(\boldsymbol{x}^{\mathrm{a}}, \mathbf{P}^{\mathrm{a}})_i$. The distribution of prior uncertainties $p(\mathbf{R},\mathbf{P}^{\mathrm{b}})$ is computed by analysing the likelihood of the innovation vector $p(\boldsymbol{y}^{\mathrm{o}} - \boldsymbol{x}^{\mathrm{b}}|\mathbf{R},\mathbf{P}^{\mathrm{b}},\boldsymbol{x}^{\mathrm{b}})$. The final product of the marginalized inversion is the node of the aggregated pdf $(\boldsymbol{x}^{\mathrm{a}})_i$ and its associated covariance matrix $\mathbf{P}^{\mathrm{a}}$. The implementation of the method is divided into three main steps to derive the optimal posterior state of emissions and the associated uncertainties.

First, the node of $p(\mathbf{R},\mathbf{P}^{\mathrm{b}})$ is obtained from the maximum likelihood computed with a pseudo-Newtonian algorithm. This couple $(\mathbf{R},\mathbf{P}^{\mathrm{b}})^{\mathrm{opt}}$ would actually give the $\boldsymbol{x}^{\mathrm{a}}$ corresponding to the node of the posterior pdf $p(\boldsymbol{x}|\boldsymbol{y}^{\mathrm{o}},\boldsymbol{x}^{\mathrm{b}})$ but with too small posterior uncertainties. Therefore, in a second step, a Monte-Carlo ensemble on $p(\mathbf{R},\mathbf{P}^{\mathrm{b}})$ is used to get a sample of the whole distribution of $p(\boldsymbol{x}|\boldsymbol{y}^{\mathrm{o}},\boldsymbol{x}^{\mathrm{b}})$, as illustrated in Figure 1. In the last step, the final $\mathbf{P}^{\mathrm{a}}$ is deduced from the shape of the distribution. The method being based on Monte-Carlo estimates of the posterior distribution, the computational costs should be tightly controlled. This is done by limiting the detectable spatial and temporal resolutions of posterior fluxes in space and time. The expert-knowledge required on the covariance matrices in the classical method is then partially transferred to the definition of the resolutions of the components of the state vector (described in Section 3.3). The relevancy of these choices may be checked a posteriori by examining the posterior error covariances (see Section 2.3).

When computing the maximum likelihood, emissions which are not constrained enough are filtered out to avoid generating numerical artefacts on top of aggregation errors. These under-constrained fluxes are detected with the influence matrix, $\mathbf{KH}$




(defined by Cardinali et al., 2004), available at each step of the computation. The diagonal terms of this matrix are between 0 and 1 and represent the sensitivity of each component of $x$ to the inversion. When the algorithm reaches a local minimum, the fluxes for which the sensitivity is less than 0.5 are filtered out (Berchet et al., 2015a).

5     Large gradients in the concentrations, which are due to emission hot-spots, are an issue. Peaks in the emissions generate fine plumes (in space and time) that the transport model may not be able to simulate accurately. The detection of such plumes is based on the diagonal terms in $(\mathbf{R}, \mathbf{P}^{\mathrm{b}})$ following a highly-skewed pdf at the end of the maximum likelihood (Berchet, 2014). All observations for which the uncertainty is in the largest 5% of $\mathbf{R}$ are filtered out; all regions of emissions for which the uncertainty is more than 500% are also filtered out. With this filtering, observations influenced by "hot-spots" of emissions are not assimilated and regions seen only through plumes are not inverted.

## 10   2.3   Tools for the interpretation of the results

This analytical method with Monte-Carlo ensemble gives access to quantifying tools, which help to better understand the influence of the various information sources within the inversion.

### Prior uncertainty

The prior fluxes are provided by yearly inventories (Section 3.4) and their uncertainties are computed from our marginalization 15 (Section 2.2). For unconstrained components (for example, emission regions that never influence concentrations at any measurement sites), prior uncertainties cannot be obtained. Therefore, the uncertainty for these components is computed based on the mean of the covariances of constrained components. The final prior uncertainty then includes prior uncertainties for both constrained and unconstrained components. This uncertainty represents the atmospheric point of view, i.e. it estimates how well the prior fluxes enable the model to reproduce the signal in the atmospheric concentrations. It is therefore higher when the 20 difference between simulated concentrations and the data is larger. In the following, it is called $\sigma_{prior}$.

### Posterior fluxes and uncertainties

The posterior fluxes $x^{\mathrm{a}}$ and their uncertainty matrix $\mathbf{P}^{\mathrm{a}}$ are determined from the Monte-Carlo ensemble of $(x^{\mathrm{a}}, \mathbf{P}^{\mathrm{a}})_i$ (Section 2.2). As the distribution of $(x^{\mathrm{a}})_i$ is symmetric relative to its node, we compute $x^{\mathrm{a}}$ as the median of the Monte-Carlo samples: $x^{\mathrm{a}} = median(x_i^{\mathrm{a}})$. The posterior uncertainties and correlations of errors are defined by the covariance matrix of the 25 ensemble $(x^{\mathrm{a}})_i$. Correlations are used to analyse the temporal and spatial structure of the constraints on the fluxes provided by the observation network. The posterior uncertainty is obtained from the tolerance interval covering 68.27% of the Monte-Carlo ensemble of posterior state vectors $(x^{\mathrm{a}})_i$. This uncertainty is then equivalent to the one-sigma interval in a Gaussian case and hereafter written $\sigma_{post}$.





**Temporal and spatial scales informed by the inverse system**

The method provides the full posterior error covariance matrix $\mathbf{P}^a$ (i.e. not only its diagonal terms). It is possible to use the correlations in $\mathbf{P}^a$ to determine which components of the state vector can be considered independent (in time and/or space) from one another by the inversion. Due to atmospheric mixing and the limited number of observations, the inversion may

meet difficulties in separating some regions. This is generally indicated by low uncertainty reduction for these regions and high positive or negative correlations between them. Here we use the correlations of errors to group blocks of emissions (see Sect. 4.1) as a conservative proxy for the temporal and spatial scales constrained by the inversion.

**Constrained fluxes and influence of the observation sites**

The influence matrix $\mathbf{KH}$ gives the constraints on the fluxes. By de-aggregating the influence according to the prior fluxes, and

taking into account the correlations, the distributed constraints on the fluxes are obtained. They may be expected to be linked to the intensity of emissions and to the distance to the stations.

The sensitivity matrix $\mathbf{HK}$ gives the sensitivity of the inversion to a change in one component of the observation vector. An observation with a high sensitivity brings strong constraints on the inversion. The weight of each station in the inversion can be computed by summing up the corresponding diagonal elements of $\mathbf{HK}$.

**Building inferred fluxes**

As stated before, all fluxes are not constrained by the inversion because some fluxes do not have any significant impact on the observations. Also, some inverted fluxes may not be robust enough (see Section 4.2). To build total fluxes, we then use the posterior emissions when available and robust, and the prior emissions otherwise (see Section 4.2 and Section 4.3.1). The obtained fluxes are called hereafter inferred fluxes (they are not the same as the posterior fluxes which result directly from

the inversion). The uncertainty on inferred fluxes is computed from the prior and posterior uncertainties by assuming that the posterior and prior parts are independent from each other and calculated as follows:

$$\sigma_{inferred} = \sqrt{\sigma_{post}^2 + \sigma_{prior}^2} \tag{4}$$

**Error reduction**

The final error reduction, after post-processing of the Monte-Carlo outputs, brought by assimilating the atmospheric data may

be estimated with:

$$R = \left(1 - \frac{\sigma_{inferred}}{\sigma_{prior}}\right) \times 100. \tag{5}$$

**3 Inversion set-ups**

For this study, we use the domain-limited chemistry-transport model CHIMERE at $10 \times 10$ km$^2$ over France (Section 3.1) and focus on the year 2012 for which four stations provided CH$_4$ measurements in France and five in the neighbouring countries





(Section 3.2). Two inversions are performed: one called "regional run" and the other "sectorial run". The regional run aims at estimating the total $CH_4$ fluxes by region. It consists of using geographical areas, defined so that the size of the problem is reasonable but each area is physically consistent and aggregation errors are assumed to be small. The sectorial run focuses on the national $CH_4$ emissions by sectors. It consists of using the various sectors for methane sources available in the prior

(Section 3.4) and assuming that each type of source is consistent enough over the whole country to be inverted as a whole. As a results the state vector is defined differently for the two runs (Section 3.3).

The inferred fluxes of $CH_4$ for 2012 are obtained from a series of 12 monthly inversions. In the following, the inversion set-up is given for one month, the 12 monthly inversions having been run independently both for the regional run and the sectorial run.

## 3.1   Observation operator: CHIMERE model

The chemistry-transport model CHIMERE is an area-limited 3D Eulerian chemistry-transport model (`www.lmd.polytech-nique.fr/chimere/`; Menut et al., 2013), embedded in the inversion system PYMAI developed at LSCE (Berchet et al., 2015a; Berchet, 2014). The full description of CHIMERE and references are available in Menut et al. (2013). The area of interest in our study is mainland France, at a horizontal resolution of $10 \times 10 \text{ km}^2$. Boundary conditions are interpolated from global

simulations (see Section 3.4 for details). To limit the aggregation errors due to the coarse resolution of boundary conditions, a buffer region around mainland France is defined with intermediate horizontal resolutions (Figure 2). With this grid, the global coarse information on concentrations is only used at the scale of the hemispheric background, while neighbouring regions are explicitly included in our simulations focussing on mainland France. On the vertical, 29 levels are defined from the surface to 300 hPa, with a finer resolution close to the surface (first levels at $\approx$ 5, 40, 85, 135 m a.g.l then geometrical increase).

The model is forced by the European Centre for Medium-range Weather Forecast (ECMWF) forecast at 12 hours, available every 3 hours, interpolated at $0.15° \times 0.15°$. The relevant fields (horizontal wind, temperature, humidity, etc.) are then interpolated hourly on the horizontal and vertical grid of CHIMERE. The transport schemes are of order 1 on the vertical and 2 on the horizontal; deep convection is taken into account with Tiedke's scheme.

For each component of the state vector $x^b$ (see Section 3.3), response functions (i.e. the contributions of this component to

the simulated concentrations equivalent $\mathbf{H}x^b$ to the observation data $y^o$) are computed. The 200 (for the regional run) or 136 (for the sectorial run) simulations are then summed up.

## 3.2   Observation vector

In 2012, measurements of atmospheric $CH_4$ mixing ratios were available at 4 stations in France and 5 stations in the neighbouring countries, mainly north from France (Figure 3). Their coordinates are given in Table 1. Hourly means of continuous data

are all reported on the same scale (NOAA2004). The measurements are made mostly by optical instruments, such as Picarro or Caribou instruments and by gas chromatographs at GIF and PUY (Lopez et al., 2015; Schmidt et al., 2014; Yver-Kwok et al., 2015). Taking into account failures and maintenance of the instruments, data are not available during the whole year, as indicated in Table 1 and on the time-series in the supplementary material (Sections S1 and S3).



Since our problem is to be explicitly solved, the size of the error covariance matrix for observations, **R**, must be small enough. Moreover, the observation data used must be consistent with the space and time resolutions chosen for the problem. Therefore, we used hourly means (provided with the associated variance) computed from the continuous measurements. When several levels are available at a site, only the highest one is retained since the transport model is not always able to optimally

represent vertical mixing close to the surface.

Among the available data (Table 1), we used hourly means in the afternoon (defined as the period from 14 h (included) to 19 h (not included) UTC) only when the boundary layer height (BLH) is higher than 500 m in the model (selected data displayed in Sections S1 and S3). This choice is made to avoid periods when the representation of vertical mixing in the model is not adapted for atmospheric inversion (Vautard et al., 2009).

The spatial distribution of the stations is not homogeneous throughout France: stations are sparse in the most western part of the country and in the South-east. The time coverage is also heterogeneous and sometimes sparse (e.g. BIS, Table 1 and Fig. S.1). Heterogeneous sampling of atmospheric concentrations may influence the performance of the inversion, which is further discussed in Section 4.2.

### 3.3 State vectors

For each monthly run, the fluxes are optimized at the weekly scale: 3 weeks of 8 days and a last "week" of 5 to 7 days depending on the month, leading to a number of components of 4 times the number of regions or sectors. The lateral boundary conditions are adjusted every two days (or three days at the end of 31-day months) for each of the 4 lateral borders and the top of the domain, leading to 75 (70 in February) components. The initial methane concentrations are adjusted by one coefficient for the whole 3D concentration field at the first time-step.

For the regional run, the French regions were delimited based on the land-use and vegetation type, according to GlobCover v2.3 (Defourny et al., 2011) and ECOCLIMAP (Champeaux et al., 2005). Limiting the size of the problem and according to the two aforementioned maps, we chose to define 26 regions in France. Four other regions were added to represent the neighbouring continental areas and a last one for the sea. The 31 regions are represented in Figure 3.

As a result, for the regional run, the state vector for one month has 200 components:

– 1 component for initial conditions

    – 75 components for boundary conditions (only 70 components in February).

    – 124 components for emissions (i.e. 31 regions during 4 "weeks").

For the sectorial run, we use the SNAP (Selected Nomenclature for Air Pollution) sectors from 1 to 10 for anthropogenic $CH_4$ emissions (see Table 2 for the definition of the sectors). Other sources are neglected (including natural emissions such as

from wetlands and termites). $CH_4$ emissions are split into SNAP sectors over France only. For the neighbouring continental regions and the sea, total emissions are used.

As a result, for the sectorial run, the state vector for one month has 136 components:





- – 1 component for initial conditions

- – 75 components for boundary conditions (only 70 components in February).

- – 40 components for emissions in France (i.e. the 10 SNAP sectors during 4 "weeks")

- – 20 components for emissions in the 5 outlying areas (continental areas A to D and sea E in Figure 3) during 4 "weeks".

For each component, the propagation to compute the response function is 6 days (the domain is supposed to be ventilated after this delay).

### 3.4  Prior information

**Initial and boundary conditions**

For the initial and boundary conditions in our domain, $CH_4$ optimized concentrations at the global scale for 2010 by Bousquet
et al. (2006) are used. The initial spatial resolution of the 3D-fields was $3.75° \times 2.5°$, longitude and latitude respectively, with 19 vertical levels from the surface to the stratosphere. A time resolution of 48 hours was used. These concentration fields were spatially and temporally interpolated to our model resolution (Section 3.1). Even though 2012 was not available at the time of our study, using 2010 values ensures that the large scale variations at the boundaries are realistic in terms of seasonal cycle. The impact on the final results of using 2010 values instead of 2012 is small since boundary conditions are optimized in the
inversion (see Section 4.2).

**Methane emissions**

Emission estimates used as prior knowledge of $CH_4$ fluxes are taken from the European annual anthropogenic emission inventory produced by IER (Institut für Energiewirtschaft und Rationelle Energieanwendung Universität Stuttgart) for 2005 (Pregger et al., 2007). This inventory estimates French mainland annual $CH_4$ emissions at 3108 Gg $CH_4$. Emissions are provided for
10 SNAP sectors, the main emitting sectors in France being agriculture (SNAP10, about two thirds of the total anthropogenic emissions) and waste treatment and disposal (SNAP9, about 17% of the total anthropogenic emissions, Table 2). SNAP5 (non-industrial combustion plants) contributes ≈3.5% of the total anthropogenic emissions, and SNAP2 (distribution of fossil fuel) about3%. These four SNAP sectors represent a total of 99% of the prior emissions. SNAP6 (solvents and other products) does not emit $CH_4$. Sources other than those included in these 10 sectors are neglected, including natural emissions such as
from wetlands since their total area (and contribution to atmospheric concentrations) were assumed to be small in France. This assumption will be further discussed in Section 4 when discussing the French methane yearly budget. The choice of a larger scale anthropogenic inventory has been made because the CITEPA does not provide gridded emissions and the INS was not available at the time of this study. Forward sensitivity tests have shown that IER was the inventory ensuring the best performances over France in simulating $CH_4$ concentrations at stations compared to the global scale inventory EDGAR. The
EDGARv4.2FT2012 inventory (EDGAR 4, 2009) estimates larger $CH_4$ emissions over France (3866 Gg $CH_4$ in 2012) and leads to larger discrepancies between observations and forward simulations.





The IER $CH_4$ inventory is available at a 10 minute horizontal resolution (about 15 km) for each SNAP sector. The emission maps were interpolated on the grid of the model with an hourly time-resolution. The total emission map used as the prior is shown in Figure 4, the emissions maps for each sector are presented in the supplementary material (Section S2).

## 4    Results and discussion

One of the main objectives of this study is to assess $CH_4$ emissions using atmospheric data at the yearly scale and to compare with bottom-up estimates. With our method we can determine the components of the state vector that are actually constrained in the inversion. This allows us to define the spatial and temporal scales that are resolved by our system (Sect. 4.1) and to determine how each station constrains the system and which regions or sectors are constrained (Sect. 4.2). The inferred fluxes are reconstructed from the posterior estimates of the constrained components and the prior estimates for the un-constrained ones: this is first done at the monthly scale (Sect. 4.3.1) to discuss seasonal variations (Sect. 4.3.2), and finally at the yearly scale (Sect. 4.3.3) to compare our top-down estimate with bottom-up ones.

### 4.1    Space and time scales resolved by the inversion

Assessing the spatial and temporal scales resolved by an inversion system is critical for establishing future network design strategies and correctly analysing the outputs of the inversion. As detailed in Section 2.3, the posterior error covariance matrix $\mathbf{P}^a$ is used to assess which spatial and temporal scales are solved by the inversion. Components of the state vector are considered to be actually separated by the inversion when the associated correlations in the posterior error covariance matrix $\mathbf{P}^a$ are lower than a given threshold (see Section 2.3). In the regional run, the threshold must be set so as to avoid over-interpreting spatial information; in the sectorial run, the threshold must be set so as to avoid unduly separating sectors. In the following, a "block" is a set of components that are considered correlated together (i.e. a group of components among which the correlations are all higher than the chosen threshold). A given block may include emissions for various weeks and various regions/sectors together with initial conditions and boundary conditions. A high correlation between fluxes and boundary conditions may be due to over-corrections of emissions to create a background concentration signal: a few ppbs of error in boundary conditions can be compensated by non-realistic increments on fluxes inside the domain; conversely, an error on emissions in the buffer regions can be compensated by non-realistic increments on boundary conditions. This is why, in the following, we discard such increments by taking into account only blocks including exclusively emissions (neither initial nor boundary conditions).

The correlation threshold must be set at a value that avoids two issues (as explained by Berchet et al., 2014, 2013): too high a threshold leads to always separating all the components ($\geq 0.7$, Figure 5 b), which implies a high risk of over-interpreting small scale results since patterns of corrections forming dipoles in neighbouring regions are not grouped; whereas a lower value leads to large blocks of regions covering half of France ($\leq 0.3$, Figure 5 c). For this study, the correlation threshold is set at a balanced value of 0.5, which gives the largest number of blocks of more than one component (Figure 5 a) as well as a small residual correlation between the blocks (Figure 5 d) and the second smallest mean area covered by one block (Figure 5 c). In the regional run, this mean block area corresponds almost to the finest available spatial resolution for emissions in the



state vector, $\approx 68{,}000$ km$^2$. The same threshold is set for the sectorial run, which gives the second largest number of blocks of more than one component (Figure 5 a).

The components of interest correspond to the 26 French regions (numbered 1 to 26 in Figure 3) in the regional run or to the 10 SNAP sectors in the sectorial run. With a correlation threshold of 0.5, in the regional run, 260 components of interest are seen over the year, among 1248 weekly components (26 French regions × 4 "weeks" × 12 months), and about 55% of these 260 are correlated at least to another one. In the sectorial run, 92 components of interest are seen over the year, among 480 components (10 sectors× 4 "weeks" × 12 months), and about 35% of these are in a block with at least another one.

The components that are seen and grouped indicate that the regional spatial resolution with 26 regions is neither too coarse (individual regions are seen) nor too fine (some regions are grouped together) from the atmospheric point of view compared to the information that can be retrieved from the atmospheric data into the emission space. More measurement sites would allow the inversion to constrain emissions at a finer spatial resolution.

The weekly time resolution seems to be close to the finest resolution at which the inversion is actually informative. The components corresponding to a given region through the 4 weeks of a month are almost never grouped (2 cases of 2 weeks in the same group among the 64 groups). Running inversions with a coarser time resolution (e.g. bi-weekly or monthly), in the state vector would therefore be equivalent to assuming perfect correlations between weeks, which are not suggested by the information in the atmospheric signal. Nevertheless, using results at the weekly scale would lead to a risk of over-interpreting the time-windows when a posterior is available compared to the weeks not seen by the inversion. Finally, the best compromise to interpret the results of the inversion is to aggregate them at a coarser time resolution (monthly and yearly), as described in Section 4.3.1. Moreover, the yearly time scale is, *in fine*, the one that top-down approaches have to target to be integrated as control methods that check the national emission reports and their trends, in order to meet societal and political needs.

## 4.2 Constrained areas and sectors

By de-aggregating the influence matrix according to the prior fluxes, the constraints on the fluxes are obtained (see Section 2.3). The constraint at a given time and location then depends both on how well a source is detected in the atmospheric signal and on the intensity of the flux. It is a good indicator of the efficiency of the inversion since there is not much interest in having information on an area where the emissions are known to be small or null. In Figure 6, the total annual constraints on regions independent from initial and boundary conditions (see Section 4.1) are displayed for the year 2012 together with the average weights of the stations (computed from the sensitivity matrix, see Section 2.3). These weights are displayed on a scale with arbitrary units traceable to degrees of freedom of the signal. For instance, BIS contributes more to the constraints than CBW in the regional run (Figure 6, see Section S3 for details of the whole year at each site).

As the domain covers neighbouring areas as well as France itself, stations outside France (CBW, CRP, MHD, RGL, TAC) can help constraining fluxes outside of France and the boundary conditions. To quantify the impact of these stations on the constraints on the fluxes in France, a regional run was carried out without them. The total annual sum of constraints on French CH$_4$ fluxes in the regional run without these outside-France stations is more than 1.7 times smaller than the constraints provided in the reference regional run assimilating data from all available stations. When using the stations outside France, the influence





of the components for fluxes outside France and the boundary conditions is partly taken into account by the information provided by the stations outside France. The information provided by stations located in France is then more efficiently used for constraining French fluxes, the influence of outside fluxes and boundary conditions being otherwise taken out from the atmospheric signal.

As expected, the regional run shows that most areas where stations are sparse are not well constrained. Thus, the South-east of France is not very well constrained in 2012, more specifically regions in the Alps (18, 19 in Figure 3) and close to the Mediterranean coast (9, 26, 13 in Figure 3). The Pyrenees (14 in Figure 3) are not constrained at all, as well as regions in the East (e.g. 1, 11 in Figure 3). Newly operated stations in Germany or the South-east of France can thus be expected to improve our spatial coverage of French $CH_4$ emissions. Nevertheless, the best constrained regions are not necessarily those

where measurement sites are located. Indeed, regions 3, 4, 5 (numbers in Figure 3) in the West, are better constrained than regions 22 and 23 (between OPE and GIF) and region 15 (close to PUY). The spatial distribution of constraints actually depends on the intensity of fluxes and of the distance to stations. The best constrained fluxes are not necessarily the closest to the stations because plume situations are filtered out in the inversion (see end of Section 2.2). The best constrained fluxes are then in areas upwind of the stations at distances between 100 and 300 km, when plumes are spread out and the atmospheric

signal is smoothed enough to be compared with the transport model. As a result, Brittany (regions 3 and 5 in Figure 3) is well constrained (Figure 6 c) although it is far from the stations, because the prior fluxes are among the most intense (10–20 g.m$^{-2}$, Figure 6 b) and the western circulation brings well-mixed air masses from this region to GIF. On the other hand, PUY does not always constrain the regions closest to the station (15 and 20 in Figure 3) very well: the local transport brings filtered-out plumes from local emissions when the station is in the boundary layer and clean air masses (containing almost no information

on French surface fluxes) when the station is in the free troposphere. The region close to BIS is not well constrained as the wind comes either from the Atlantic ocean, with no influence from the French emissions, or from the East, with either relatively small fluxes (0.5–2g.m$^{-2}$, Figure 6 b) or local plumes from nearby towns directly impacting the station and then being filtered out.

In the sectorial run, the four major contributors to methane emissions, SNAP10, SNAP9, SNAP2 and SNAP5 are constrained

(Table 2, Figure 7). The other sectors are never seen by the inversion: the constraints are null.

## 4.3   National emissions

### 4.3.1   Reconstruction of inferred emissions and error reduction

In the following sections, inferred estimates of the French emissions are based on the posterior fluxes, where and when fluxes are constrained. Where and when fluxes are not constrained, the values of prior fluxes are used (see Section 4.2) to reconstruct

the inferred estimates of French emissions.

In the regional run, from February to December, between 28 and 65% of the national monthly prior fluxes are constrained (Table 3); fluxes in January are less constrained (14%), which may be linked to the smaller number of individual observations available after selection (348 against more than 430 for the other months). In the sectorial run, between 45 and 94% of the





national monthly total prior fluxes are constrained, apart from February (14%) and May (38%), as detailed in Table 4. As explained in Section 4.2, these constrained fluxes belong to the four most emitting sectors, SNAPs 2, 5, 9 and 10, representing 99% of the total prior emissions (Table 2). The differences in the constrained fraction of emissions between the two runs are due to the different resolutions. A sector covering the whole of mainland France may be constrained by any one of the

available stations; conversely, if no data is available (e.g. all are filtered out because of plume situations), the whole sector is not constrained.

Since the inferred emissions are built from a patchwork of posterior and prior fluxes, the differences between the prior and the inferred emissions are larger where constraints are stronger, as displayed in Figure 4. Both runs agree on the main patterns of correction applied to the prior emissions, with smaller fluxes around Paris and larger fluxes in Normandy and Brittany

(regions 3, 4 and 5 in the regional run) as well as in the Centre (regions 15 and 8). Not surprisingly, the regional run infers more contrasted fluxes than the sectorial run. Indeed, the regional run can optimize regions separately and eventually create contrasts, while the sectorial run keeps the (smoother) prior distribution of each sector, which is scaled for the whole of France. Such a difference is clearly visible in the Centre of France (Figure 4, middle panel). The positive corrections are due to SNAP10 (agriculture), which is also the sector with the largest emissions. The negative corrections around Paris are due to SNAP9

(waste treatment and disposal) and, for a smaller part, SNAP2 (non-industrial combustion plants) and SNAP5 (distribution of fossil fuels).

At the monthly scale, the uncertainty on inferred fluxes is smaller than on the prior (Figure 8 a) for both runs. In the regional run, the monthly error reductions (computed as explained in Section 2.3, Table 3) on national budgets are larger than 25% (up to 72%, median at 39%) with the exception of January (≈17%), when only 14% of the fluxes are constrained (see above).

In the sectorial run, the error reductions are larger than 25% for 8 months (from 37 to 90%); for the 4 remaining months (February, April, May and June), for which less than 50% of the fluxes are constrained, the error reductions are smaller than 16% (Table 4).

### 4.3.2 Seasonal variations

In both runs, from a constant prior, the inferred fluxes vary over the year with larger emissions during the summer (June to

August for the regional run, July and August for the sectorial run, Figure 8 a). The amplitude of the monthly variations of the inferred median fluxes are ≈260 Gg $CH_4$ in the regional run and ≈265 Gg $CH_4$ in the sectorial run (Figure 8 a). Generally, both runs are statistically compatible, i.e. the inferred confidence ranges overlap, with the exception of September and December. A similar seasonal variability was found by the inversions in the InGOS project (Bergamaschi et al., 2017): among the 4 systems providing monthly variations, 3 have a maximum in August, with amplitudes of ≈130 to 170 Gg $CH_4$ over the year

(Figure 8 c). The variations introduced by the inversion may be an artefact due the variations in the number of assimilated data. Nevertheless, the consistency between the two runs, which use the same data but for constraining different state vectors, and with the inversions in the InGOS project, which do not use the same set-up and data, strongly suggests that the inferred variations are due to actual characteristics of the fluxes. In this case, the variations introduced by the inversion may be due

(c) Author(s) 2017. CC BY 4.0 License.





to natural sources (which are not included in our prior) and/or to seasonal variations in anthropogenic sources, which are not taken into account in the yearly inventories.

Natural sources of $CH_4$ in France are assumed to originate mainly from natural wetlands or termites. Other natural emissions involve lakes and the natural out-gassing of the Earth and are hardly quantified at the moment at this scale, but are expected

neither to be large nor to bring significant contribution to the seasonal cycle of methane emissions. Natural wetland emissions in France have been estimated from several vegetation models in the framework of an international inter-comparison project (11 models; Poulter et al., 2016) at $200\pm150$ Gg $CH_4.y^{-1}$ with a peak-to-peak amplitude of 15–35 Gg $CH_4$. The peak season is in September-October (which may correspond to accelerated methanogenesis under warmer temperatures and larger amounts of labile substrates) and the smallest emissions occur in February-March. This contribution of wetlands therefore cannot explain

by itself the inferred seasonal variations in our total emissions. Emissions by termites are not expected to vary much over the year, though information is missing to document their variations.

Therefore, these results strongly suggest that anthropogenic sources largely contribute to the seasonal variability. The sectorial run indicates that the month-to-month variations are mainly due to agriculture (SNAP10 in Figure 8 b). Indeed, since most of French $CH_4$ emissions are due to agriculture (75% according to our prior, Table 2 SNAP 10), which intensity varies during the

year (generation of agricultural waste, sensitivity of microbial decomposition to temperature and humidity), seasonal variations in this sector may actually be large. Nevertheless, the actual period of maximum/minimum emissions is not easy to assess in the inventories. For example, $CH_4$ emissions from cattle are linked to several parameters, including the age and activity of the animal (e.g., in France, Vermorel, 1997; Vermorel et al., 2008); similarly, in Switzerland, Henne et al. (2016) indicate that the transhumance of cows is not taken into account in the inventories. Emissions from waste treatment and disposal (SNAP 9),

particularly water waste treatment, also display seasonal variability (Spokas et al., 2011).

Overall, the inferred seasonal variations are likely to be due to agricultural (and for a smaller part, waste) emissions super-imposed with contributions of the natural sources, which the inversion has had to attribute to one of the available sectors since natural sources were not included in the prior emissions and no new sector could be created by the inversion.

### 4.3.3 Yearly budget

Our study estimates total yearly $CH_4$ emissions in France to be 3835–4051 Gg $CH_4$ based on the regional run and 3570–4193 Gg $CH_4$ based on the sectorial run (Table 5). As mentioned previously, these two runs are consistent at the yearly scale.

Our results are also statistically consistent (i.e. the inferred confidence ranges overlap) with those derived from the set of atmospheric inversion systems participating in InGOS (Bergamaschi et al., 2015b, Table 5 or Figure 9 "Total"). The range provided by InGOS is computed from the differences between average values from the various systems and not, as in our

study, from an analysis of the errors. If the uncertainty of each system was taken into account, the range for InGOS would be larger still. A comprehensive inter-comparison of inversion methods and systems with a common data set should be considered at the national scale as it is done at the continental scale in the framework of InGOS.

The atmospheric inversions of French emissions (our study and InGOS) consistently suggest that $CH_4$ emissions may be up to two times larger than the estimates provided by anthropogenic inventories (Table 5). As stated in Section 3.4, the nat-





ural emissions were not included in the prior emissions. These natural emissions are estimated at $200\pm150$ Gg $CH_4.y^{-1}$ for wetlands and 209 Gg $CH_4.y^{-1}$ for termites, i.e. 10-15% of anthropogenic French emissions. In the future, when finer spatial resolution maps of wetland emissions will be available, these natural emissions should be included to better represent the prior knowledge of the emissions at the French national scale. Taking into account these known estimates of natural emissions, the

median values of the inferred emissions by top-down approaches (our study and InGOS) are still systematically larger than the total estimates provided by bottom-up approaches (any anthropogenic inventory added to wetland and termite emissions, Figure 9 and Table 5). Our inferred $CH_4$ emissions are about 25 to 55% larger than bottom-up estimates (median values in Table 5). For example, our atmospheric inversions lead to $CH_4$ emissions about 35% larger than the most recent anthropogenic inventory dedicated to France, INS, summed-up with the median estimate of natural emissions; the CITEPA median estimate

(reported to UNFCCC), added to the median natural source estimates, is about 35% smaller than our estimates.

The partitioning between emission sectors is available for the sectorial run and most of the inventories (Figure 9). Since the natural emissions have to be attributed to an already defined sector, we chose to assume that most of them were attributed to SNAP10. This assumption is mainly based on the fact that the spatial distribution of agriculture makes it the most consistent with the spatial distribution of natural emissions. Indeed, the other sectors seen by the inversion (SNAP2, 5 and 9) are

not diffuse enough to match the patterns of natural emissions by wetlands or termites (Section S2). Also, the atmospheric inversion attributes about 84% of the total emissions to agriculture (Figure 9 "SNAP10"), while agriculture emissions from inventories added to natural emissions from wetlands and termites represent 68–79% of the total bottom-up estimates (Figure 9). Assuming the natural emissions are included in SNAP10 in the sectorial run, the posterior estimate for these sources is 2970–3580 Gg $CH_4$, i.e. about 66% and 18% larger than the agriculture emissions by INS and IER, respectively, plus natural

emissions.

Emissions due to waste treatment and disposal (SNAP9) are reduced by the inversions and estimated at only 380–460 Gg $CH_4$ in the sectorial run compared to 657 Gg $CH_4$ in the INS. SNAP9 inferred emissions are lower than any bottom-up median estimates, except ECLIPSE.

Emissions by the distribution of fossil fuels (SNAP5) are estimated at 81–155 Gg $CH_4$, on the higher range of the bottom-up

estimates (23–155 Gg $CH_4$). From the atmospheric inversions, the relative uncertainty on the SNAP5 emissions (about 30%) is expected to be large since these emissions are very localized in areas where natural gas distribution systems are built and operated, and, as such, might not always be seen by the inversion, especially after our filtering of hotspots (see Fig. S.13).

Finally, emissions by the residential sector (SNAP2, non-industrial combustion plants) stay very close to the prior by IER, mainly because it is not strongly constrained (see Section 4.2 and Figure 7 c).

Top-down estimates, from our study and the InGOS project, are in agreement. They both find larger $CH_4$ emissions in France than the bottom-up methods (inventories and biogeochemical models). Moreover, in our study, the filtering out of hot-spots limits the risk of over-estimating the posterior emissions due to the assimilation of a few high concentration peaks. Therefore, the atmospheric inversions hint at an under-estimation of French $CH_4$ emissions in the inventories. The possible under-estimation of $CH_4$ emissions in the bottom-up methods could be due to an underestimation of the emission factors or



activity data, or to underestimations resulting from extrapolation/interpolation procedures in the anthropogenic inventories, or to an underestimation of the natural sources (including other natural sources than wetlands and termites).

## 5 Conclusions

In this study, we have inferred $CH_4$ emissions in mainland France in 2012 by assimilating continuous atmospheric mixing
ratios measurements from the European network ICOS into a Bayesian inversion system. Two runs were performed in order to use the atmospheric information in different ways: one case is based on regions of emissions i.e. a view in terms of correcting the spatial distribution of fluxes and the other is based on emission sectors i.e. a view in terms of source activities.

The analytical method we used allows us to compute several diagnostics and to derive insights on the strengths and limitations of our set-up, in a consistent statistical approach. The first issue is to assess which spatio-temporal scales are actually
constrained by a relatively sparse network in a country with large regional variations in emissions. Our results show that, with a network of four continuous stations inside France and five in the neighbouring countries, regions about 50,000 km$^2$ and a time resolution of about one week are close to the finest resolutions at which information can be retrieved from the available atmospheric data into the emission space.

The network providing continuous atmospheric mixing ratio data was set up as a European infrastructure. Therefore, the ques-
tion arises of the constraints it can bring on emissions at the national scale. As expected given the relatively small number of measurement sites and their heterogeneous spatial distribution, regions where stations were sparse in 2012 were not well constrained, i.e. particularly in the South-east of France. This limitation could now be overcome as two stations have been set up in the Observatoire de Haute-Provence and at the Cap Corse in 2013 and 2014. Further work is needed to quantitatively estimate their impact but they will certainly contribute to better constrain the fluxes in the South-east and Corsica. Other stations outside
France are also now available, in Spain, Italy, Switzerland and Germany.

From the quantitative diagnostics derived from the analytical method, we decided to exploit the results of our inversions at the monthly and yearly scale for the regional and sectorial inversions. These results are ranges of emissions, equivalent to a one-sigma interval in a Gaussian framework.

The monthly totals reveal seasonal variations of French methane emissions in 2012. Both our inversions are statistically
consistent (i.e. the inferred confidence ranges overlap) with each other most of the year (10 months out of 12). The uncertainties are large ($\pm 166$ to $173$ Gg $CH_4$) in May and June for activity sectors, because of agriculture and, possibly, natural emissions. We assume that natural emissions have mostly been attributed by the inversion to the agriculture sector because its spatial distribution is the closest to the diffuse pattern of natural fluxes. The seasonal variations we find are consistent with other inversions from the InGOS project, with a maximum in summer (July–August) and a peak magnitude of about 260 Gg $CH_4$.
We assumed that the consistency with various inversion set-ups makes it likely that this seasonal signal is not an artefact due to the varying number of assimilated data. These seasonal variations may indeed be due to actual variations in the agricultural (and for a smaller part, waste) emissions superimposed with variations in the natural sources, but cannot be explained by natural sources alone, considering the biogeochemical model estimates for wetland emissions used in this study.



Our estimated $CH_4$ emissions for France in 2012 range from 3835 to 4050 Gg $CH_4$ and from 3570 to 4190 Gg $CH_4$ for the regional run and the sectorial run, respectively. Our two runs are statistically consistent with each other and also with the InGOS results of a set of top-down studies based on different chemistry-transport models and inverse systems. To compare our estimates with bottom-up estimates, we added the emissions reported by inventories dedicated to anthropogenic emis-

sions with natural emissions from wetlands and termites computed from biogeochemical models. Our atmospheric inversions inferred total $CH_4$ emissions about 25 to 55% higher than bottom-up estimates. In the sectorial run, for instance, inferred agriculture emissions are increased by 18% compared to the prior, leading to agriculture emissions up to 66% larger than the lowest bottom-up estimates (by the CITEPA).

In our study, the filtering out of high concentration peaks (in plume situations) limits the risk of over-estimating the posterior

emissions. Therefore, the possible under-estimation of $CH_4$ emissions in the bottom-up approaches needs to be further investigated. First, it would be useful to assess the potential origin of such an underestimation in the anthropogenic inventories (in terms of emission factors, activity data or extrapolation/interpolation procedures); second, it would be needed to better assess natural sources of $CH_4$ at the national scale.

The main differences between the prior bottom-up emissions and the inferred emissions are *(i)* smaller fluxes around Paris,

mainly due to waste treatment and disposal and to a lesser extent to non-industrial combustion plants, and, *(ii)* larger fluxes in Normandy and Brittany as well as in the Centre, because of agriculture and, possibly, natural fluxes (wetlands and termites).

The uncertainties on our total annual budgets are $\pm 108$ and $\pm 312$ Gg $CH_4$, respectively, which is smaller than the range of variation of the available inventories (from 2689 to 3666 i.e. $\pm 488$ Gg $CH_4$, anthropogenic and natural values added). The uncertainties on the fluxes by activity sectors could probably be decreased with information from isotopic data or other

source-specific tracers (such as ethane for the gas and oil sector).

Further steps of this work include runs with additional observations, method improvement and extension to other species. The building up of the ICOS network should allow us to better constrain the different regions and refine the results in the upcoming years. The main methodological improvement would be to assimilate more data each day so as to make better use of the available continuous mixing ratio measurements. In this study, night-time data and data acquired when the boundary

layer height is small are filtered-out whereas they contain the strongest signals due to regional emissions. Cautious integration of such data should increase our confidence in inferred local emissions. Finally, the PYMAI-CHIMERE inversion system will have to be adapted for the inversions of $CO_2$ and $N_2O$ fluxes at the national scale.

*Acknowledgements.* This work has been supported by the GMES-MDD program (Global Monitoring for Environment and Security-Ministère du Développement Durable) by the French ministry of sustainable development. The study extensively relies on the meteorological data pro-

vided by the ECMWF. Calculations were performed using the resources of LSCE, maintained by F. Marabelle and the LSCE IT team. We also wish to thank Simona Castaldi and Monia Santini for providing methane emissions from termites to the Global Methane Budget project. We are grateful to the modellers who provided estimates of methane emissions from wetlands under the umbrella of the Global Methane Budget project: Charles Koven, Xiyan Xu, and William Riley for CLM4.5, Joe Melton and Vivek Arora for CTEM, Hanquin Tian for DLEM, Thomas Kleinen for LPJ–MPI, Ben Poulter and Zhen Zhang for LPJ–wsl, Renato Spahni and Fortunat Joos for LPX-Bern, Sushi Peng for



ORCHIDEE, David Beerling, Peter O. sHopcroft, Lila Taylor and David J. Wilson for SDGVM, Zhu Qiuan for TRIPLEX and Akihiko Ito and Makoto Saito for VISIT. We acknowledge Peter Bergamaschi for sharing InGOS results, and the inverse modellers who participated in the InGOS project for estimating European methane emissions: Peter Bergamaschi for TM5, Ute Karsten for TM3-STILT, Aki Tsuruta for TM5-CTE and Alistair J. Manning for NAME. The funding of Irish data is from the Irish Environmental Protection Agency; TAC and RGL

5    are funded by the UK Department of Business, Energy and Industrial Strategy (formerly the Department of Energy and Climate Change).





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



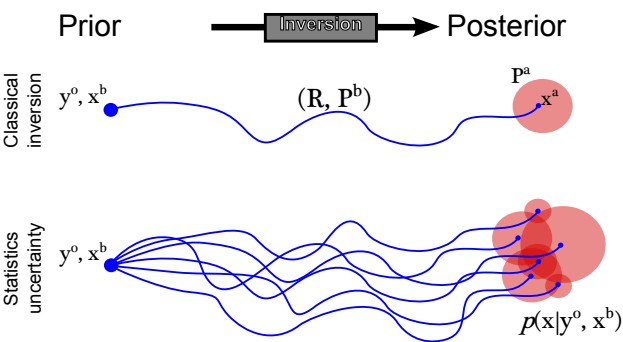

**Figure 1.** *Statistic uncertainty in Bayesian inversion. The inversion infers the posterior state $\boldsymbol{x}^a$ from $\boldsymbol{y}^o$ and $\boldsymbol{x}^b$. In the classical Bayesian framework, $\boldsymbol{x}^a$ is inferred together with its uncertainty $\mathbf{P}^a$ from the covariance matrices $(\mathbf{R}, \mathbf{P}^b)$ (top). To account for uncertainties on the error statistics, an ensemble of $(\mathbf{R}, \mathbf{P}^b)$ couples can be tested to infer an ensemble of $(\boldsymbol{x}^a, \mathbf{P}^a)$ (bottom) which are part of $p(\boldsymbol{x}|\boldsymbol{y}^o, \boldsymbol{x}^b)$.*

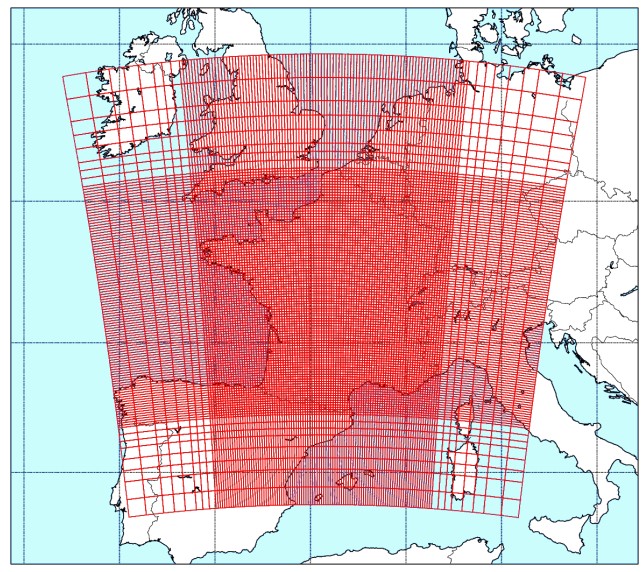

**Figure 2.** *Horizontal grid used by CHIMERE (see Section 3.1). Resolution in the centre (mainland France): 10 km×10 km for 98×98 grid cells. The sizes of grid cells increase in areas not covering mainland France: 30, 50 and 80 km over 3, 3 and 2 rows of grid cells.*



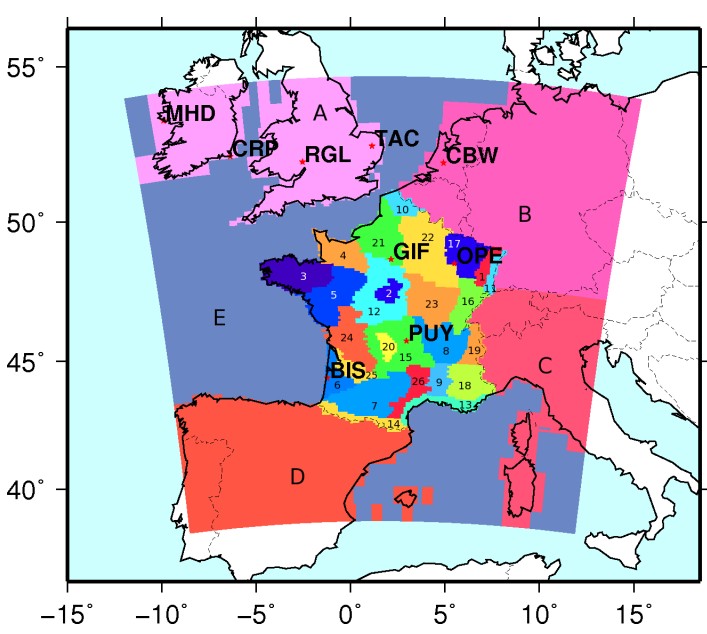

**Figure 3.** *Colors: regions for emissions, 26 regions in France (numbers), 4 "outside" regions (letters A to D) and 1 sea region (E). Stars and names: sites at which measurements were available in 2012 for CH$_4$, see characteristics in Table 1.*





**Figure 4.** *Annual median CH$_4$ emission fluxes in gCH$_4$/m$^2$ in France (top, prior from IER, inferred fluxes from the regional and sectorial runs); differences inferred minus prior (middle) for the regional and sectorial runs; details for the sectorial run (bottom): differences inferred minus prior for the 4 sectors which are actually seen, SNAPs 2, 5, 9 and 10.*





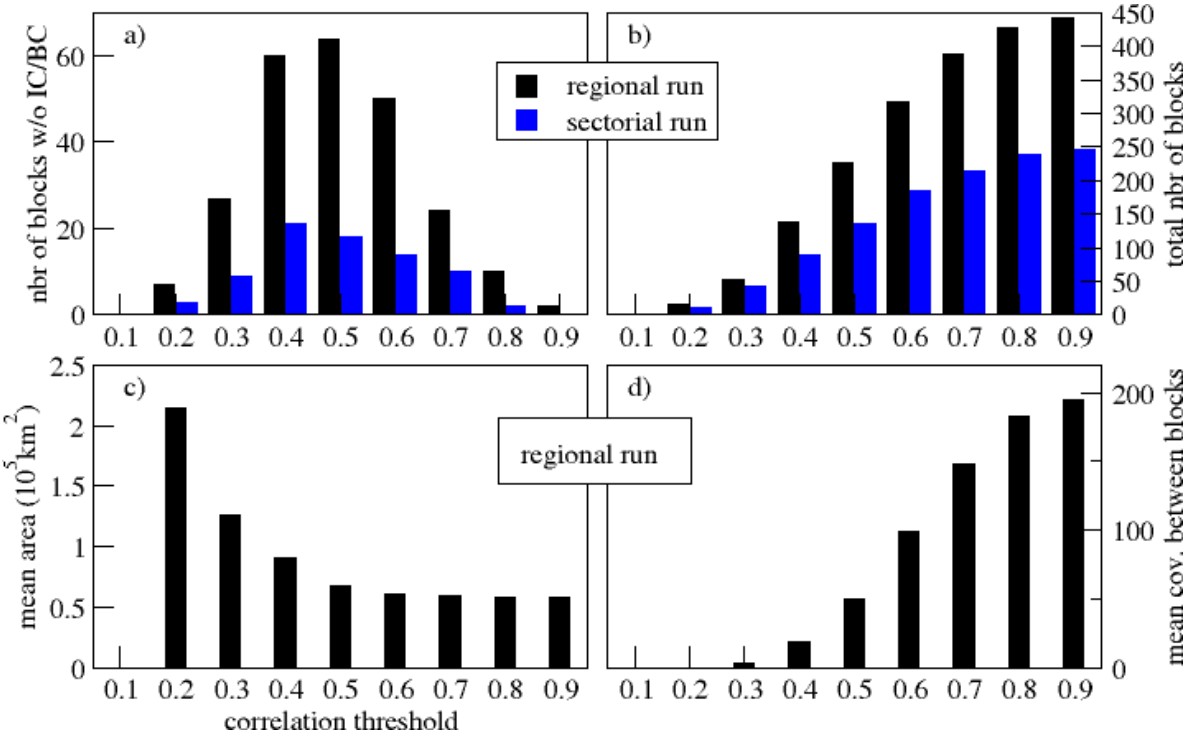

**Figure 5.** *a) Annual number of blocks of at least two components independent from both initial conditions (IC) and boundary conditions (BC) for various correlation thresholds for the regional (black) and the sectorial (blue) runs. b) Annual total number of blocks (i.e. including blocks of only one region also, compared to a) independent of IC and BC. The larger the correlation threshold is, the larger the total number of blocks is and the smaller the number of blocks of at least two regions, since less regions are considered correlated together. c) Annual mean area covered by a block for the regional run. d) Annual mean covariance between blocks for the regional run.*





**Figure 6.** *Constrained areas in the regional run (described in Section 3 and Section 3.3). The influence matrix (a, influence for each grid cell given in % over the whole year) is de-aggregated according to prior fluxes (b) to obtain the constraints (c): the annual sum of constraints on CH$_4$ emissions by the atmospheric data is shown on a logarithmic scale (right, adimensional). Red is for a strong constraint. The spatial resolution is the grid of the model (see Figure 2). Only fluxes independent from initial and boundary conditions are used (see Section 4.1). Black bold lines show the borders of the regions; grey regions are never constrained. The relative contributions of the stations in the inversion, averaged over the year are shown on an arbitrary scale (d), white being for a small contribution.*





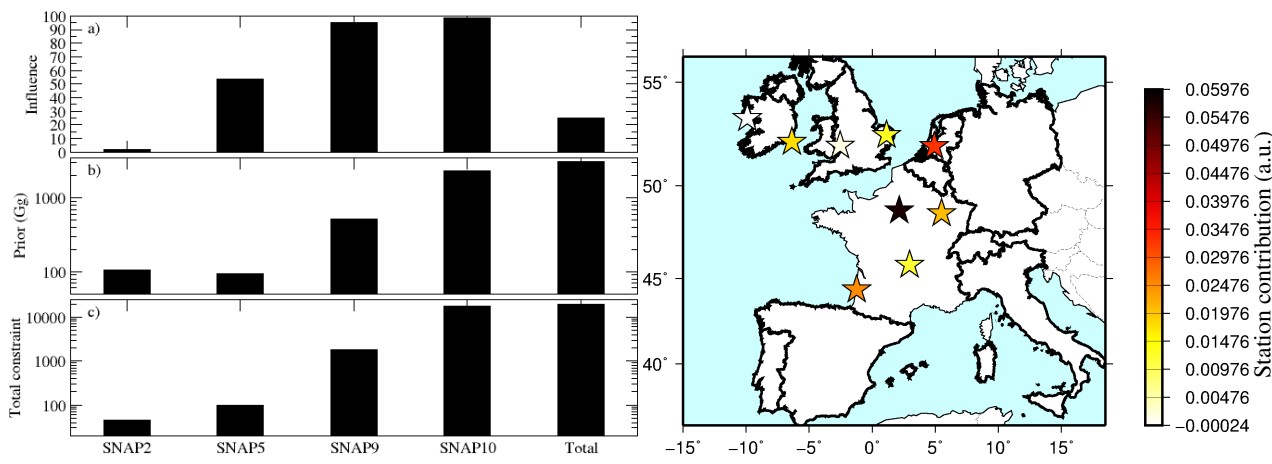

**Figure 7.** *Constraints obtained for the sectorial run (described in Section 3 and Section 3.3). The influence matrix (a, influence given in % for the whole domain over the whole year) is de-aggregated according to prior fluxes (b) to obtain the constraints (c): the annual sum of constraints on CH$_4$ emissions in the whole domain by the atmospheric data is shown on a logarithmic scale (adimensional). Only fluxes independent from initial and boundary conditions are used (see Section 4.1). Only the sectors which are actually seen are displayed. The relative contributions of the stations in the inversion, averaged over the year are shown on an arbitrary scale (map), white being for a small contribution .*



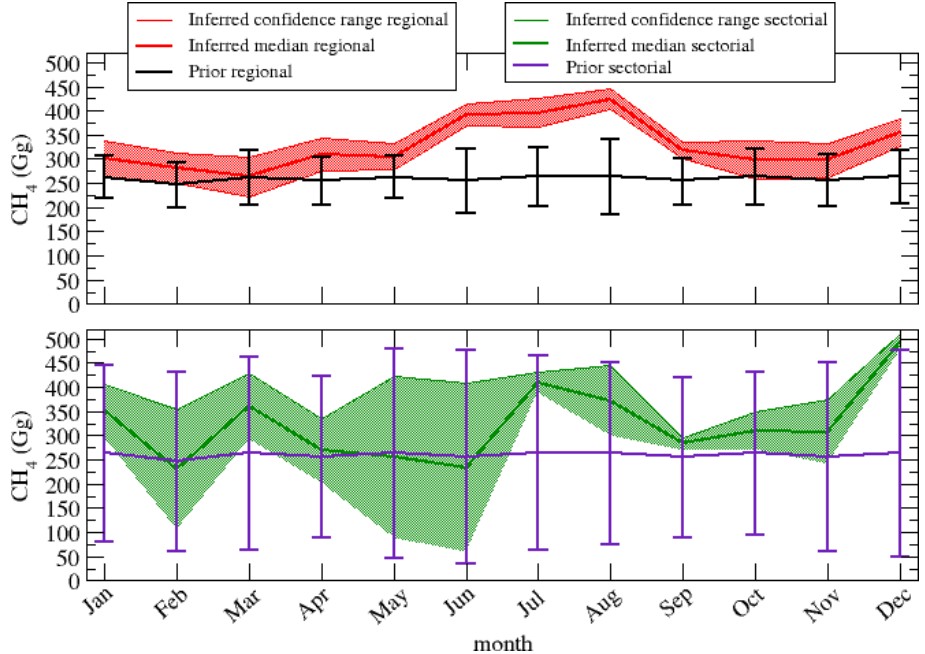

(a) Prior and inferred emissions, regional and sectorial runs

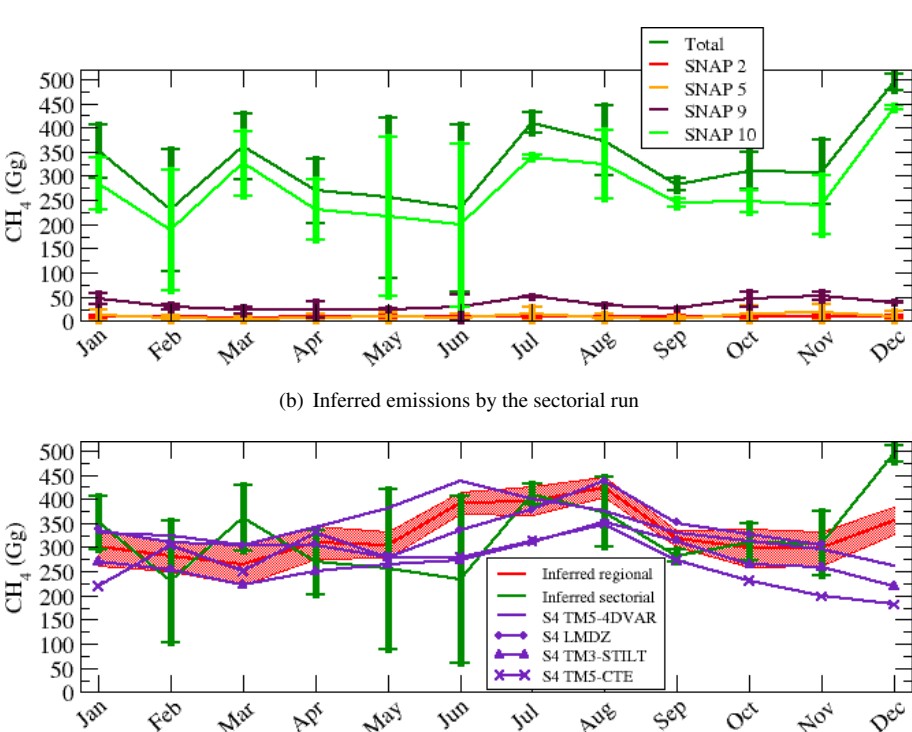

(b) Inferred emissions by the sectorial run

(c) Inferred emissions by both our runs compared to the InGOS project

**Figure 8.** *$CH_4$ monthly emissions (in Gg $CH_4$) in France in 2012 by the regional and sectorial runs. a) Prior fluxes (provided as detailed in Section 3.4) with the uncertainty computed by our method (Section 2.3) and confidence range of inferred fluxes with the median shown as a solid line (Section 2.3, Section 4.2). b) For the sectorial run: details of inferred monthly emissions for the four SNAPs which are actually seen by the inversion. c) Comparison of both runs to the inversions "S4" in the InGOS project for which monthly emissions are available (Bergamaschi et al., 2017).*



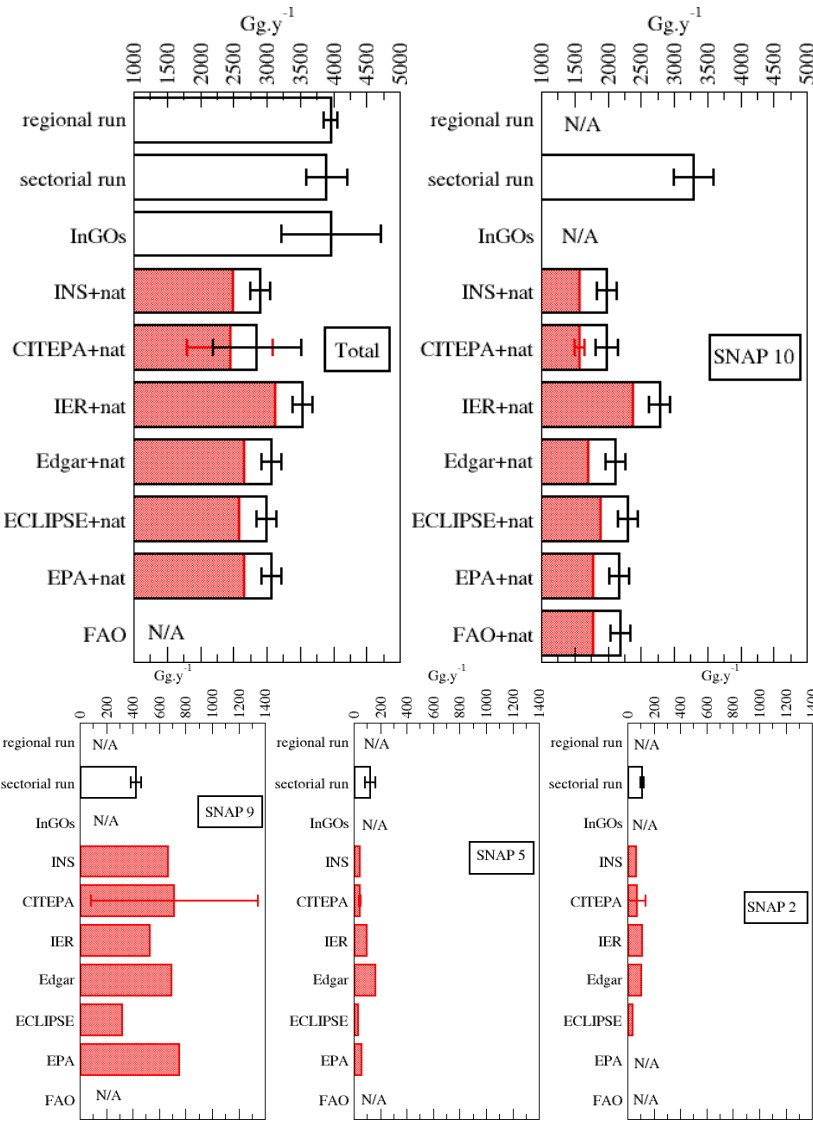

**Figure 9.** *CH₄ yearly emissions (in Gg CH₄) in France for this study and the other studies/inventories listed in Table 5. "Total": the inventories including only anthropogenic emissions (in red) are summed-up with the natural emissions by wetlands and termites. For these totals (black bars), the error bars (in black) are obtained from the range on wetland emissions (Table 5) combined with the uncertainty on anthropogenic emissions (in red), when available i.e. only for CITEPA (Table 5). The CITEPA provides uncertainties only for the main emitting sectors so that the error bar on the total emissions is under-estimated. "SNAP10": the inventories including only anthropogenic emissions (in red) are summed-up with the natural emissions by wetlands and termites. Only the CITEPA provides an uncertainty (in red), which is combined to the range on wetland emissions to obtain the error bar on the whole sector (in black). Other sectors: only the CITEPA provides an uncertainty for these sectors. N/A = not-available or the definition of sectors/activities does not match those of SNAPs.*





**Table 1.** *Characteristics of the available stations at the time of the study (see map in Figure 3). The altitude is above sea level; the height is above ground level. The total number of available data is the number of hourly means available for the whole year (i.e. maximum 8784). The number of selected data is the number of hourly means available from 14 h (included) to 19 h (not included) UTC when the boundary layer height is higher than 500 m in the model. The time coverage is computed over the afternoon hours ([14 h-18 h]) i.e. 100% for 1830 hours. At PUY, two different instruments measure $CH_4$.*

| Station | Name | Altitude (m a.s.l) | Height of inlet (m a.g.l) | Total number of available data | Number of selected data | Time coverage in 2012 (% of 1830 hours) |
|---------|------|---------|---------|---------|---------|---------|
| BIS | Biscarrosse | 120 | 47 | 2976 | 339 | 19 |
| CBW | Cabauw | 0 | 200 | 5213 | 682 | 37 |
| CRP | Carnsore Point | 9 | 14 | 7764 | 1116 | 61 |
| GIF | Gif-sur-Yvette | 160 | 7 | 7013 | 1072 | 59 |
| MHD | Mace Head | 8 | 24 | 4240 | 518 | 28 |
| OPE | ANDRA | 390 | 120 | 7384 | 1041 | 57 |
| PUY | Puy-de-Dôme | 1465 | 10 | 7037+6132 | 1036+933 | 57 + 51 |
| RGL | Ridge Hill | 199 | 90 | 6511 | 928 | 51 |
| TAC | Tacolneston | 56 | 100 | 3729 | 521 | 28 |



**Table 2.** *Prior yearly total methane emissions (in Gg CH$_4$) in France from IER interpolated on the model's grid; the crosses indicate sectors which are constrained by the atmospheric inversion (in the sectorial run).*

| SNAP | Description | Gg CH$_4$ | % of the total | Constrained |
|---|---|---|---|---|
| 1 | combustion in energy and transformation industries | 2 | 0.1 | |
| 2 | non-industrial combustion plants | 107 | 3.4 | x |
| 3 | combustion in manufacturing industry | 2 | 0.1 | |
| 4 | production processes | 2 | 0.1 | |
| 5 | distribution of fossil fuel and geothermal energy | 94 | 3.0 | x |
| 6 | solvents and other product use | 0 | 0 | |
| 7 | road transport | 21 | 0.7 | |
| 8 | other mobiles sources and machinery | 2 | 0.1 | |
| 9 | waste treatment and disposal | 522 | 16.8 | x |
| 10 | agriculture | 2356 | 75.8 | x |
| Total | | 3108 | | |

**Table 3.** *Regional run: French monthly total CH$_4$ emissions (in Gg CH$_4$) in 2012: prior confidence range (provided by our method, see Section 2.3), fraction of prior constrained by the inversion in %, confidence range for the inferred emissions and error reduction in % (see Section 2.3 for definition).*

| Month | Prior (Gg CH$_4$) | Fraction constrained (%) | Inferred (Gg CH$_4$) | Error reduction (%) |
|---|---|---|---|---|
| January | 218 - 308 | 14 | 263 - 338 | 17 |
| February | 200 - 292 | 28 | 251 - 313 | 32 |
| March | 206 - 319 | 37 | 223 - 304 | 28 |
| April | 205 - 304 | 40 | 275 - 343 | 32 |
| May | 219 - 306 | 28 | 279 - 332 | 39 |
| June | 188 - 321 | 43 | 369 - 414 | 66 |
| July | 203 - 323 | 37 | 367 - 426 | 51 |
| August | 186 - 340 | 65 | 403 - 446 | 72 |
| September | 208 - 302 | 59 | 301 - 336 | 62 |
| October | 204 - 322 | 41 | 260 - 339 | 33 |
| November | 201 - 309 | 28 | 262 - 334 | 33 |
| December | 207 - 319 | 35 | 328 - 383 | 51 |





**Table 4.** *Sectorial run: French monthly total CH$_4$ emissions (in Gg CH$_4$) in 2012: prior confidence range (provided by our method, see Section 2.3), fraction of prior constrained by the inversion in %, confidence range for the inferred emissions and error reduction in % (see Section 2.3 for definition).*

| Month | Prior (Gg CH$_4$) | Fraction constrained (%) | Inferred (Gg CH$_4$) | Error reduction (%) |
|---|---|---|---|---|
| January | 161 - 366 | 65 | 295 - 407 | 45 |
| February | 120 - 372 | 14 | 104 - 355 | 1 |
| March | 125 - 401 | 52 | 293 - 429 | 51 |
| April | 175 - 334 | 45 | 203 - 336 | 16 |
| May | 93 - 434 | 38 | 90 - 422 | 3 |
| June | 67 - 442 | 50 | 61 - 407 | 8 |
| July | 124 - 402 | 93 | 389 - 430 | 85 |
| August | 148 - 378 | 56 | 301 - 445 | 37 |
| September | 176 - 333 | 75 | 271 - 295 | 85 |
| October | 190 - 336 | 51 | 273 - 349 | 48 |
| November | 119 - 391 | 53 | 241 - 375 | 51 |
| December | 98 - 429 | 94 | 478 - 512 | 90 |



**Table 5.** *Estimates of yearly total CH$_4$ emissions (in Gg CH$_4$) in France: top-down for our study and the European project InGOS (result from 6 different models), bottom-up for anthropogenic inventories, 11 biogeochemical models for natural fluxes from wetlands and 1 model for emissions by termites. Some methods do not provide uncertainties.*

| Type of flux | Area of focus | Source | Estimate (Gg CH$_4$) | Year |
|---|---|---|---|---|
| Net total | France | this study, regional run | 3835-4051 | 2012 |
| Net total | France | this study, sectorial run | 3570-4193 | 2012 |
| Net total | Europe | InGOS [a] | 3200-4700 | 2012 |
| Anthropogenic | France | INS [1] | 2469 | 2012 |
| | France | CITEPA [2] | 2430±637 | 2012 |
| | Europe | IER [*] | 3107 | 2005 |
| | world | Edgar4.3.2 [3] | 2651 | 2012 |
| | world | ECLIPSE5a [4] | 2563 | 2010 |
| | world | EPA [5] | 2650 | 2010 |
| Agriculture only | world | FAO [6] | 1760 | 2012 |
| Natural | world | Wetlands [7] | 200 [50-350] | 2000-2014 |
| | world | Termites [7] | 209 | 2012 |

[a] Bergamaschi et al. (2015b); [*] Pregger et al. (2007), also our prior; [1] Inventaire National Spatialisé Ministère de l'Environnement, de l'Énergie et de la Mer (2017); [2] CITEPA (2016), which is the reporting to UNFCCC - the values given for uncertainties are minimum since uncertainties are provided only for the main sources; [3] Janssens-Maenhout et al. (2017) ; [4] Stohl et al. (2015); [5] EPA (2012); [6] FAOSTAT: Food and Agriculture Organization of the United Nations (2017); [7] GCP-CH$_4$ Saunois et al. (2016).