# Peer review of "How a European network may help estimating methane emissions at the French national scale"

_Atmospheric Chemistry and Physics, 2017_

## Referee Comment (RC1) · Anonymous Referee #1 · 17 Nov 2017

**1   General comments**

Pison et al. present an inversion framework for estimating country scale methane emissions of France using ground-based atmospheric mixing ratio measurements. Both spatial and sectorial patterns of methane fluxes are optimized in separate inversions and compared with bottom-up and another set of CH4 inversion emissions. The manuscript is clearly written. The method is well explained and results are nicely presented.

The focus here is to identify CH4 emissions on a country scale, which is highly pertinent for the emission reports submitted by nations to UNFCCC. The other nice aspect of this research is the selective use of posterior information– that is, only when there is

an actual constraint provided by the observations. Otherwise, bottom-up emissions are left unchanged. This method is more appealing for reporting purposes as it does not sacrifice ill-constrained regions to adjust the background concentrations. In wake of concurrent debate of whether atmospheric measurement constraints should be used along with national bottom-up emissions in policy reports, the analysis and results presented here are relevant to the scientific community. I recommend publication of this manuscript in ACP after minor corrections.

**2 Specific Comments**

My only issue with this research is the lack of a spin-down period, which would have provided sufficient observational constraint for the last month of the inversion. It can be argued that as the emissions are seen by the observations within a week, due to small spatial scales, lack of spin-down should have a small impact on the emission estimates. However, the last week emissions would still be ill-constrained. In Figure 8, the large discrepancy among the inversions of this study and Bergamaschi et al. (2017) for the month of December further suggests that a lack of spin-down might be affecting the posterior emissions.

**3 Technical corrections/suggestions**

Page 3:

Line 6: "statistical information or to uncertainties" => "statistical information or due to uncertainties"

Page 10:

[Figure]

Line 7: " 2010 by Bousquet et al. 2006 are used" is confusing to the reader. Consider writing "concentrations fields optimized using the inversions setup of Bousquet et al. 2006 are used"

Page 13:

Line 25: Consider replacing the term "seen by the inversions" with something like "constrained by observations (in the inversions)".

Page 14:

Line 30: "artefact due to" => "artefact of"

Page 17:

Line 1: "data, or to" => "data, or due to" ;

Line 5: "a view in terms of correcting the spatial distribution" => "to correct the spatial distribution";

Line 11: "regions about" => "regions of about".

---

## Author Response (AR1)

**1. General comments**

Pison et al. present an inversion framework for estimating country scale methane emissions of France using ground-based atmospheric mixing ratio measurements. Both spatial and sectorial patterns of methane fluxes are optimized in separate inversions and compared with bottom-up and another set of CH4 inversion emissions. The manuscript is clearly written. The method is well explained and results are nicely presented.
The focus here is to identify CH4 emissions on a country scale, which is highly pertinent for the emission reports submitted by nations to UNFCCC. The other nice aspect of this research is the selective use of posterior information- that is, only when there is an actual constraint provided by the observations. Otherwise, bottom-up emissions are left unchanged. This method is more appealing for reporting purposes as it does not sacrifice ill-constrained regions to adjust the background concentrations. In wake of concurrent debate of whether atmospheric measurement constraints should be used along with national bottom-up emissions in policy reports, the analysis and results presented here are relevant to the scientific community. I recommend publication of this manuscript in ACP after minor corrections.

*We thank the reviewer for his/her time and for the fruitful comments to improve our manuscript. We answer below point by point to the points raised by the reviewer.*

**2. Specific Comments**

My only issue with this research is the lack of a spin-down period, which would have provided sufficient observational constraint for the last month of the inversion. It can be argued that as the emissions are seen by the observations within a week, due to small spatial scales, lack of spin-down should have a small impact on the emission estimates. However, the last week emissions would still be ill-constrained. In Figure 8, the large discrepancy among the inversions of this study and Bergamaschi et al. (2017) for the month of December further suggests that a lack of spin-down might be affecting the posterior emissions.

*We fully agree with the reviewer about the spin-down issue that can affect the quality of the last week of inversion. In this study, we have checked the results at the weekly scale to ensure that no impact due to the lack of spin-down period was easily visible in the month of December, particularly in the last week. Since the optimized emissions at the weekly scale were consistent, the impact of spin-down periods will be inquired further into in a future work. We have added a comment line 30 seq. Page 14 and Table 3: "The variations introduced by the inversion*

*may be an artefact of the variations in the number of assimilated data (number of data used per month in Table 3). Moreover, in December, the inferred peak in emissions found in both runs may be due to the limited spin-down period: data acquired till the 31st of December are used so that emissions of the last week of the month are not well constrained through having only a small impact at most stations."*

**3. Technical corrections/suggestions**

Page 3: Line 6: "statistical information or to uncertainties" → "statistical information or due to uncertainties"
*This is updated in the new version of the manuscript.*

Page 10: Line 7: " 2010 by Bousquet et al. 2006 are used" is confusing to the reader. Consider writing "concentrations fields optimized using the inversions setup of Bousquet et al. 2006 are used"
*We thank the reviewer for the suggestion to improve the clarity of the sentence, which now reads: "For the initial and boundary conditions in our domain, we use CH4 concentration fields optimized at the global scale for 2010, using the inversion set-up of Bousquet et al. (2006)."*

Page 13: Line 25: Consider replacing the term "seen by the inversions" with something like "constrained by observations (in the inversions)".
*Line 25 now reads: "All the other sectors are never inverted as constraints, provided by observations, are null."*

Page 14: Line 30: "artefact due to" → "artefact of"
*Modified in the new version of the manuscript.*

Page 17: Line 1: "data, or to" → "data, or due to" ;
*Modified in the new version of the manuscript.*

Line 5: "a view in terms of correcting the spatial distribution" → "to correct the spatial distribution";
*Lines 5-7 now read: "Two runs were performed in order to use the atmospheric information in different ways: one case is based on regions of emissions to adjust the spatial distribution of inventory-based fluxes, and the other is based on emission sectors to adjust source activities prescribed in inventories."*

Line 11: "regions about" → "regions of about".
*Modified in the new version of the manuscript.*

**2   List of all relevant changes made in the manuscript**

Numbers of pages and lines refer to the marked manuscript, to be found in the next section of this document.

- p.1, affiliation [7]: Heerhugowaard → Petten

- p.1, affiliation [a]: Dept. Phys. Geography and Ecosystem Science → ICOS ERIC - Carbon Portal

- p.1, l.9: makes → make

- p.3 l.6: or to uncertainties → or due to uncertainties

- p.4, l.30: it is usual → it is common

- p.5, l.10: here) while → here), while

- p.5, l.26: The method being based on → As the method is based on

- p.7, l.16: all fluxes are not → not all fluxes are

- p.8, l.6 : a results → a result

- p.10, l.9-10: $CH_4$ optimized concentrations at the global scale for 2010 by Bousquet et ql. (2006) are used → we use $CH_4$ concentration fields optimized at the global scale for 2010, using the inversion set-up of Bousquet et al. (2006)

- p.13, l.25: The other sectors are never seen by the inversion: the constraints are null. → The other sectors are never inverted as constraints, provided by observations, are null.

- p.14, l.30: an artefact due the → an artefact of the variations

- p.14, l30-33: the number of assimilated data → the number of assimilated data (number of data used per month in Table 3). Moreover, in December, the inferred peak in emissions found in both runs may be due to the limited spin-down period: data acquired till the 31st of December are used so that emissions of the last week of the month are not well constrained through having only a small impact at most stations.

- p.17, l.2: or to → or due to

- p. 17, l.7-8: emissions i.e. a view in terms of correcting the spatial distribution of fluxes and the other is based on emission sectors i.e. a view in terms of source activities → emissions to adjust the spatial distribution of inventory-based fluxes, and the other is based on emission sectors to adjust source activities prescribed in inventories.

- p.17, l.12: regions about → regions of about

- p.17, l.31: artefact due to the varying → artefact of the varying

[revised manuscript text omitted]